# Household structure, composition and child mortality in the unfolding antiretroviral therapy era in rural South Africa: comparative evidence from population surveillance, 2000–2015

Brian Houle ![ORCID],[1,2] Chodziwadziwa Kabudula,[2] Dickman Gareta ![ORCID] ,[3] Kobus Herbst,[3,4] Samuel J Clark[2,5]

For numbered affiliations see end of article.

**Correspondence to**
Dr Brian Houle;
brian.houle@anu.edu.au

## ABSTRACT

**Objectives** The structure and composition of the household has important influences on child mortality. However, little is known about these factors in HIV-endemic areas and how associations may change with the introduction and widespread availability of antiretroviral treatment (ART). We use comparative, longitudinal data from two demographic surveillance sites in rural South Africa (2000–2015) on mortality of children younger than 5 years (n=101 105).

**Design** We use multilevel discrete time event history analysis to estimate children's probability of dying by their matrilineal residential arrangements. We also test if associations have changed over time with ART availability.

**Setting** Rural South Africa.

**Participants** Children younger than 5 years (n=101 105).

**Results** 3603 children died between 2000 and 2015. Mortality risks differed by co-residence patterns along with different types of kin present in the household. Children in nuclear households with both parents had the lowest risk of dying compared with all other household types. Associations with kin and child mortality were moderated by parental status. Having older siblings lowered the probability of dying only for children in a household with both parents (relative risk ratio (RRR)=0.736, 95% CI (0.633 to 0.855)). Only in the later ART period was there evidence that older adult kin lowered the probability of dying for children in single parent households (RRR=0.753, 95% CI (0.664 to 0.853)).

**Conclusions** Our findings provide comparative evidence of how differential household profiles may place children at higher mortality risk. Formative research is needed to understand the role of other household kin in promoting child well-being, particularly in one-parent households that are increasingly prevalent.

## STRENGTHS AND LIMITATIONS OF THIS STUDY

⇒ We provide 16 years of prospective, harmonised population data from two rural areas of South Africa heavily impacted by HIV/AIDS to analyse associations between household structure and composition with child mortality in a unified statistical framework.
⇒ We used multilevel modelling to account for shared mortality risk for children in the same household.
⇒ We were also able to adjust for potential confounding with socioeconomic status and household head gender.
⇒ Our measure of household kin is based on household memberships, meaning we cannot account for the role of kin who are members of another household.
⇒ Our study was only able to identify the presence of matrilineal members in the household given limitations with father linkages over time.

track or have already reached this goal, the remaining global mortality burden is uneven. The majority of global under-5 deaths occur in sub-Saharan Africa.[1] A greater understanding of the social, cultural and contextual determinants of child mortality is critical to reduce child mortality in the region.

In the context of the HIV epidemic, a number of studies in sub-Saharan Africa have examined factors that impact child mortality, such as parental survival,[2–4] and HIV infection and antiretroviral therapy (ART).[5–7] Household studies have focused on the role of older household members, fostering and orphanhood.[8–12] Less attention has been paid to other contextual risk factors, especially with mortality changes associated with the HIV epidemic and more recently with the availability of ART. While there is a substantial literature from low-income and

## INTRODUCTION

Reducing preventable child mortality remains an urgent global priority and central Sustainable Development Goal. Goal 3 targets reducing under-5 mortality to 25 per 1000 live births by 2030. While many countries are on

middle-income countries on the importance of factors such as household composition and child mortality,[13–18] there remains limited longitudinal evidence from HIV-endemic areas.[3 19] The availability of ART may also have changed the relative importance of different risk factors and how they influence child mortality[6]—highlighting the need for longitudinal data examining risk factors over time.

The extent to which studies are generalisable across settings is also unclear, particularly given differences in factors such as household organisation and available resources. There is a lack of comparative research within countries, particularly among disadvantaged populations.[20] Understanding if and how household risk factors are associated with child mortality between different settings has important implications for policies and interventions to reduce preventable child mortality. Comparative evidence is needed to inform these efforts.

Our primary aim is to investigate the relationship between a child's risk of dying and their household's structure and composition. We follow the conceptual approach proposed by Madhavan et al[21] by examining

two household-level dimensions: (1) structure, referring to the configuration of generations in the household and level of nucleation; and (2) composition, referring to the availability of specific kin that may be moderated by parental presence. We use comparative, longitudinal data from 2000 to 2015 from two demographic surveillance sites (DSSs) in rural South Africa to examine these factors. The time period covers when ART was unavailable and after widespread availability, allowing for an assessment of if and how associations between different risk factors and child mortality have changed over time with ART availability.

## METHOD
### Setting
We used household census data from 2000 to 2015 for two populations living in rural South Africa. Both DSSs monitor a geographically defined population over time and collect prospective information, including demographic indicators (eg, mortality, fertility and migration) as well as household-level information and social

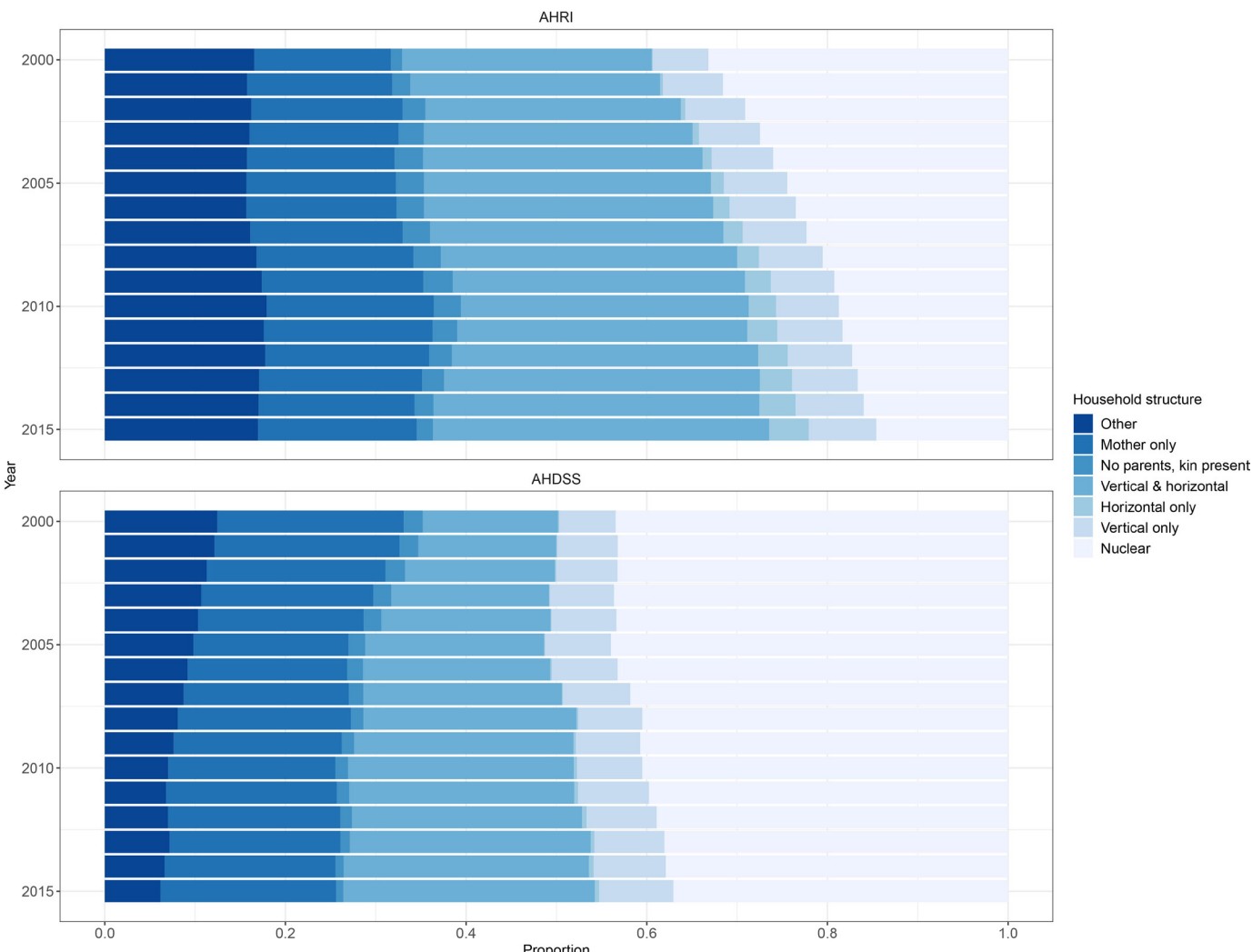

**Figure 1** Distribution of household structures over time and demographic surveillance site, Agincourt Health and socio-Demographic Surveillance System (AHDSS) and Africa Health Research Institute (AHRI), South Africa 2000–2015.

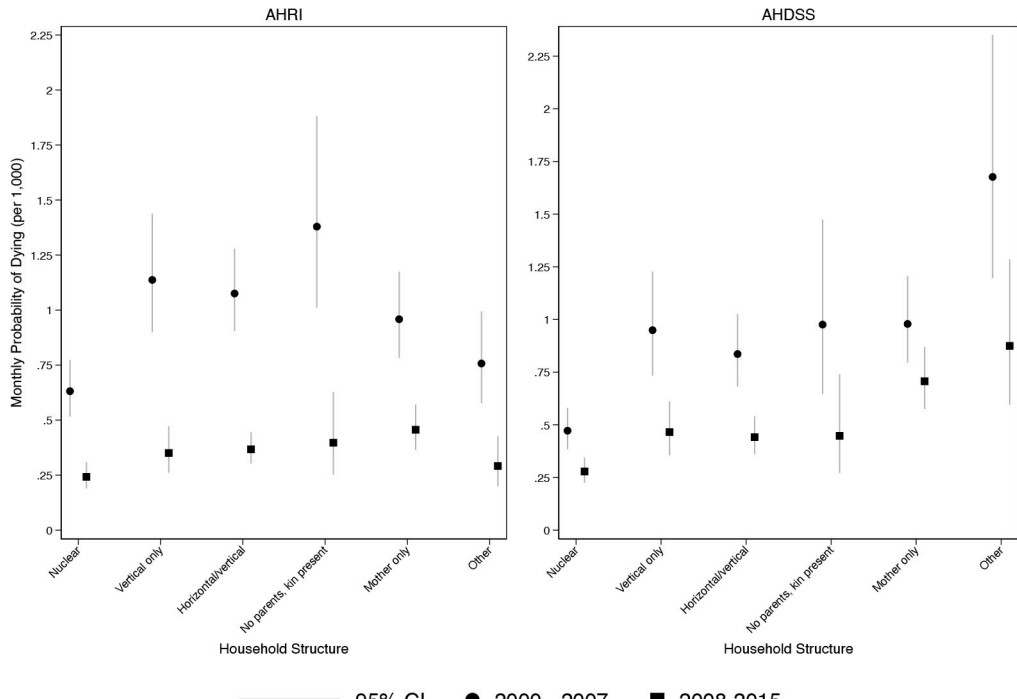

**Figure 2** Monthly probability of child death, by household structure and demographic surveillance site: Agincourt Health and socio-Demographic Surveillance System (AHDSS) and Africa Health Research Institute (AHRI), South Africa 2000–2015. Jittered points to reduce over plotting.

indicators. The first, the Agincourt Health and socio-Demographic Surveillance System (AHDSS), has been conducting an annual census update of the population since 1992, updating vital events (births, deaths and migration) along with socio-demographic information.[22] AHDSS is located in the Ehlanzeni district of Mpumalanga. The primary ethnic group is amaShangaan. The population under surveillance was approximately 90 000 people in 2011. Until recently child and adult mortality were increasing.[23] ART became available in 2008, with resulting substantial reductions in mortality.[23]

The second site, the Africa Health Research Institute (AHRI), started biannual census updates in 2000.[24] AHRI is located in the uMkhanyakude district of KwaZulu-Natal. The population is almost entirely Zulu-speaking and comprised approximately 90 000 people in 2011. HIV/AIDS has had a significant impact on the population, but the availability of ART in 2004 significantly increased life expectancy in the population.[25]

Each site conducts a verbal autopsy (VA) for individuals who died during surveillance rounds. A trained fieldworker or nurse uses a standardised VA instrument to interview the closest living relative of the decedent to record signs and symptoms experienced before their death.

### Household measures

Both DSSs update a roster of household members at each census update. To create a harmonised data structure across the sites, we used the household membership data (which included membership start and end dates)

to identify members of the same household as the index child.

Our key measures were household structure and composition. We followed the categorisations of Madhavan *et al*[21] as their analysis was based at AHDSS and included a complementary, cross-sectional data set with detailed kinship data. The structural typology included only those kin co-resident in the household: (1) both parents and no other kin (nuclear); (2) one or both parents and grandmother (vertical); (3) one or both parents, aunts/uncles (horizontal); (4) one or both parents, grandmother and aunts/uncles (vertical and horizontal); (5) no parents but any kin present; (6) mother only, no kin present; and (7) other (eg, lone father and other uncommon combinations). These categories were created based on the mother's identification number, which allowed us to describe the presence of matrilineal members in the household. Father linkages with kin are not as robust over time, given the focus of the DSSs on vital events such as fertility—our typology is therefore limited to matrilineal kin. For kinship co-residence, we created an indicator of grandmother, and counts of aunts, uncles and older brothers and sisters (aged 5+). We also included an indicator of parental co-residence as both parents, one parent or no parent resident in the household. All of the household structure and composition indicators were time-varying.

### Statistical analysis

We organised the data as person-months for children under 5 years of age (0–59 months), where each child was at risk of dying for each month they were observed (up

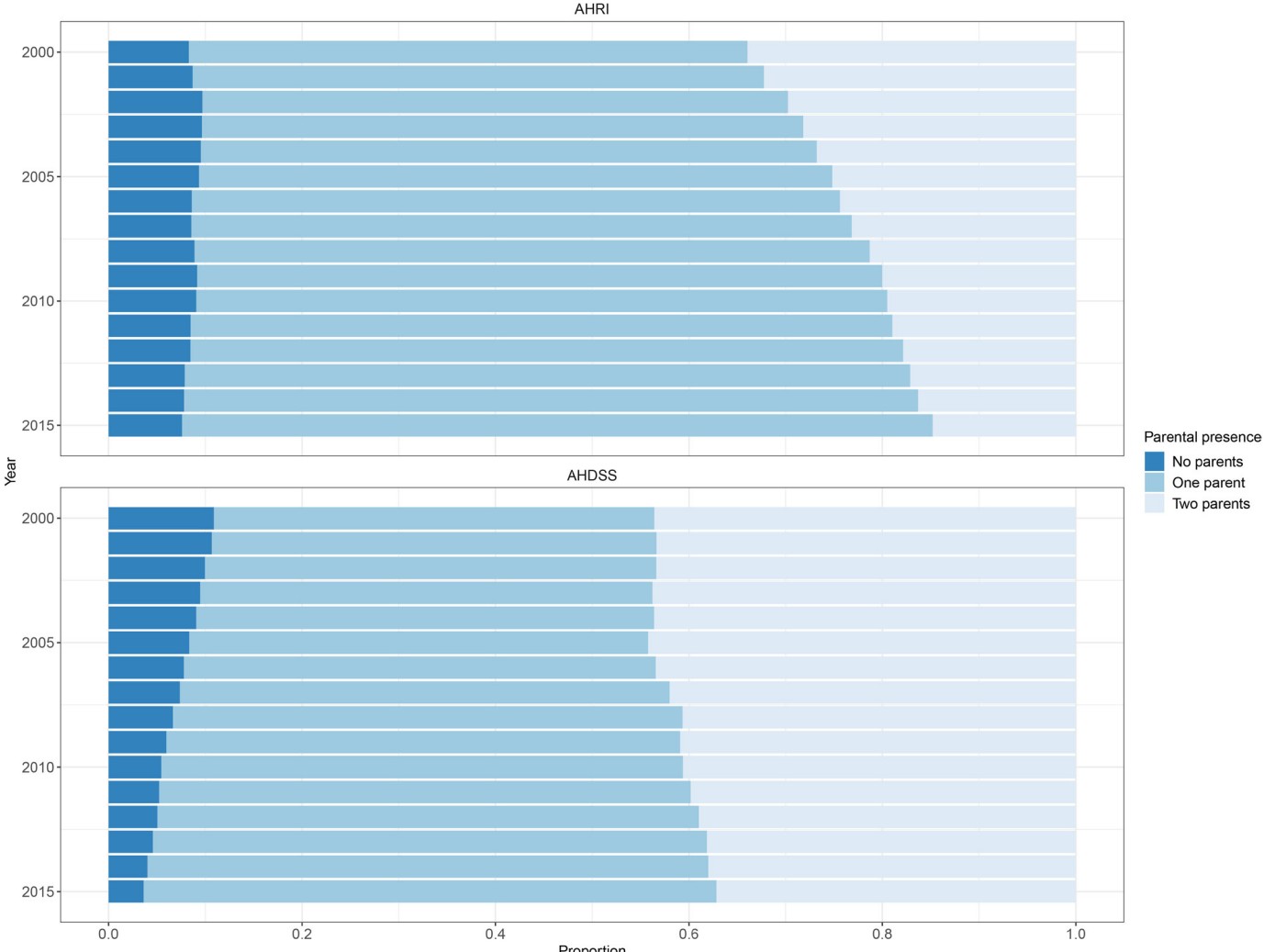

**Figure 3** Distribution of parent co-residence with children over time and demographic surveillance site, Agincourt Health and socio-Demographic Surveillance System (AHDSS) and Africa Health Research Institute (AHRI), South Africa 2000–2015.

to and including death). We modelled mortality using discrete time event history analysis and used multilevel relative risk regression models to estimate a child's risk of dying.[26 27]

To test for differences in household relationships between the two sites and over time, we included a binary site indicator (AHDSS or AHRI), and a time period indicator of 2000–2007 and 2008–2015. The split between 2007 and 2008 marks the time period when ART was available at both sites. At the household-level we included counts of the number of other household members ages 0–4, 5–19 and 20 years and older. We also included controls for child sex and age (<1 month, 1–6 months, 7–23 months and 24–59 months); multiple birth (singleton or multiple); and mother's age at birth (15–19, 20–24, 25–29, 30–34 and 35+ years). We tested if the structural typologies and kinship and parental presence associations varied over time, between sites and by child sex and age using nested likelihood ratio tests.

We used InterVA-5 to assign causes of death.[28] InterVA is a model that assigns up to three causes of death based on the VA interview data—we used the cause of death with the largest likelihood for each complete VA interview. This allowed us to compare how the distribution of causes of death changed over time within each DSS.

Finally, in a submodel we included gender of household head and household socioeconomic status (SES) as important controls for our main finding. Household SES was based on a common set of measures measured at each DSS since 2001, summarised using principal components analysis.[29] We used tertiles of the first principal component score from the most recent measurement. Given the truncated time span and higher levels of missing data on these two covariates, we used this submodel as a sensitivity test to determine if these factors explained our main findings.

**Patient and public involvement**

Neither study participants nor public were involved in study design or conduct of the study. Both AHDSS and AHRI have ongoing liaison and open dialogue with the DSS study communities and their leaders.

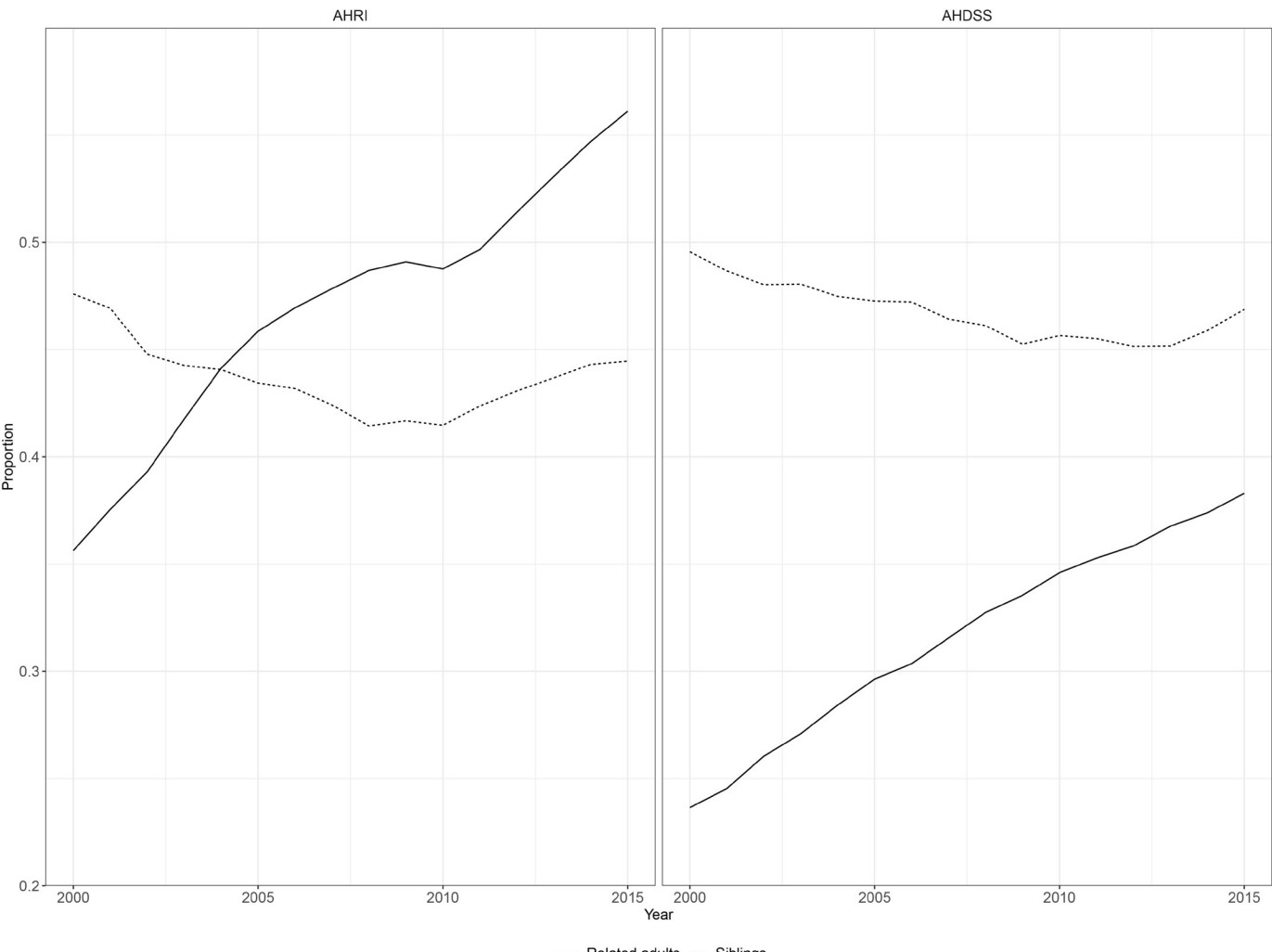

**Figure 4** Proportion of children living with related adults and siblings (not mutually exclusive) over time and demographic surveillance site, Agincourt Health and socio-Demographic Surveillance System (AHDSS) and Africa Health Research Institute (AHRI), South Africa 2000–2015.

## RESULTS

We first describe under-5 mortality patterns by time and DSS. Out of a total of 101 105 children, 3603 died between 2000 and 2015. Online supplemental figure 1 shows the mortality rate per 1000 for children under-5 by year and DSS. Mortality began decreasing at AHRI in 2004 and AHDSS in 2009. Online supplemental figure 2 shows the distribution of child cause of death over the two time periods by DSS. For both AHRI and AHDSS, the share of deaths due to HIV/AIDS and tuberculosis declined in 2008–2015 compared with 2000–2007, while the share of deaths due to respiratory infections has increased over time.

### Household structure

Next, we examine differences between DSSs in the distribution of household structures over time (figure 1). For AHDSS, about 40% of households were nuclear only compared with a declining proportion of households at AHRI over time (33% in 2000 to 15% in 2015). Vertical and horizontal households represented about one-third

of household structures at AHRI, while at AHDSS this household type has been increasing over time (15% in 2000 to 28% in 2015). Given the very low percentage of horizontal only households at AHDSS, in the subsequent mortality analysis we collapsed this group with vertical and horizontal households (initial tests indicated similar mortality patterns between these two groups). While rare at both DSSs, we preserved the no parents, kin present typology given its conceptual distinctiveness.

The results from the full household structure model are presented in online supplemental table 1. An interaction between DSS and time period (p<0.001), DSS and household structure (p=0.001) and time and household structure (p=0.004) significantly improved model fit. A multilevel model including a mother random intercept improved model fit according to the Bayesian information criterion ($\Delta BIC = 156$) and resulted in the final model. Twins, boys, younger children and children in households with two or more other children have higher mortality risk.

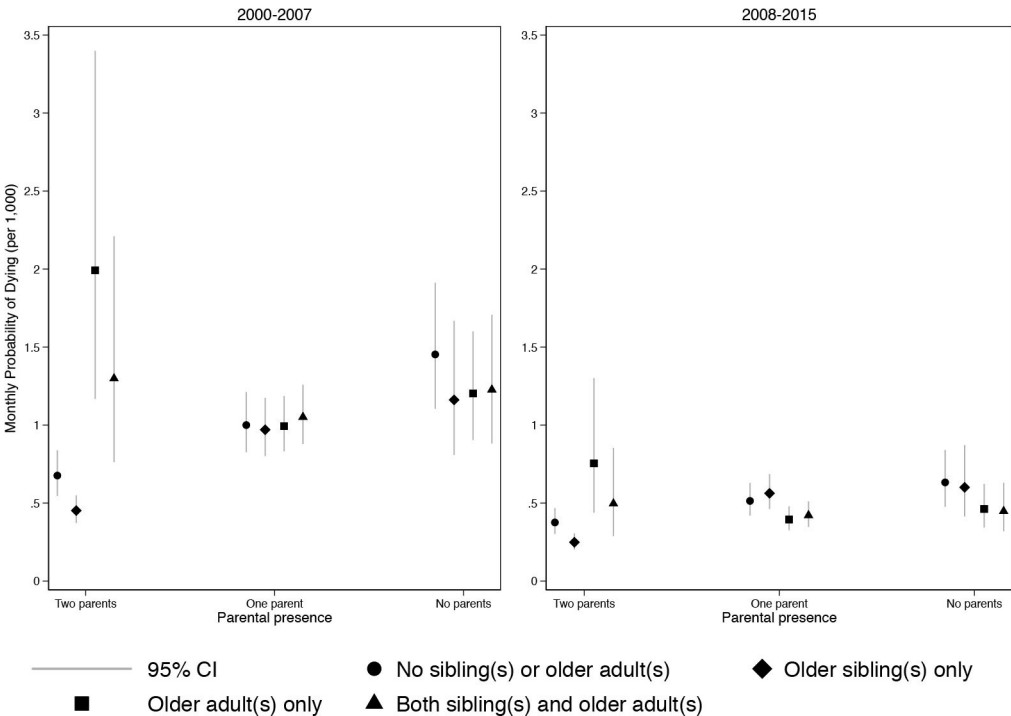

**Figure 5** Monthly probability of child death, by parent co-residence, kin presence and time period: Agincourt Health and socio-Demographic Surveillance System (AHDSS) and Africa Health Research Institute (AHRI), South Africa 2000–2015. Jittered points to reduce over plotting.

Figure 2 shows model-based predicted probabilities of a child dying by household structure, DSS and time period. For both DSSs, the period of ART availability lowered the probability of dying across the different household types. For both DSSs, children in nuclear households had the lowest probability of dying compared with other household types. While most other household types had similar probabilities of dying, AHDSS children in mother only and other household types had elevated mortality risk, even in the ART period.

### Kin presence
In developing a model for the presence of parents and kin, we tested several specifications including: counts versus binary indicators of specific types of kin and testing for differences by child sex. Because we found that only related adults and older siblings had any association with child mortality, the results of the kin presence model in online supplemental table 2 present this more parsimonious categorisation.

Figure 3 shows the proportion of children according to co-resident parents. Children living with no parents was 10% or lower at both DSSs, but has declined over time at AHDSS (11% in 2000 to 4% in 2015) while remaining relatively stable at AHRI (8% in 2000 and 2015). At both sites living with one parent was common and has steadily increased over time, but was increasingly more common at AHRI (58% in 2000 to 78% in 2015). Conversely, living with both parents remained relatively stable at AHDSS over time (44% in 2000 to 37% in 2015), but declined over time at AHRI (34% in 2000 to 15% in 2015). Comparing

with household structure (figure 1) also shows that the vast majority of children living in two-parent households did not live with other related kin.

Figure 4 shows the proportion of children living with related adults and siblings (not mutually exclusive). The proportion of children living with related adults increased over time by about 20% at AHRI and 15% at AHDSS, and was more common for AHRI children. Living with siblings remained relatively stable over time at just under 50% at both DSSs.

The results from the full parent and kin presence model are presented in online supplemental table 2. An interaction between DSS and time period (p<0.001), parent co-residence and related adults (p<0.001), parent co-residence and older siblings (p<0.001) and related adults and time period (p=0.003) significantly improved model fit. A multilevel model including a random intercept for the mother improved model fit according to the BIC ($\Delta BIC = 152$) and resulted in the final model.

Figure 5 shows the model-based predicted probabilities of a child dying by kin presence, parent co-residence and time period. Having older siblings lowered the probability of dying only for children in a household with both parents (relative risk ratio (RRR)=0.736, 95% CI (0.633 to 0.855)). While rare, having other adult kin present in two-parent households resulted in a higher probability of dying for children. Only in the later ART period was there evidence that older adult kin lowered the probability of dying in single parent households (RRR=0.753, 95% CI (0.664 to 0.853)).

## Sensitivity analysis

In the submodels including household SES and household head gender, the main findings of household structure and composition remained (see online supplemental tables 3 and 4). Children in the highest (wealthiest) SES tertile had a lower risk of dying compared with children in the lowest (poorest) tertile.

## DISCUSSION

Over a 16-year period, we examined associations between household structure and the role of kin in the household and differential mortality risk for children. Our main finding was that children in nuclear households with both parents had the lowest risk of dying. The kin presence model supported this finding, showing that having both parents as members of the household provided a protective effect for children.

The role of both parents is important to consider in light of the local context at both sites. Labour migration is common in both populations, and remittances play a beneficial role supporting children remaining in rural areas.[30] The protective role of the father has been shown in another comparative study from these sites,[2] highlighting their importance as long as they remain a breadwinner for the household.[31] However, our indicator of father presence does not capture the detail of their role and support in the household, particularly for those fathers non-co-resident in the household.[32 33] Female migration is also increasing over time,[34] with resulting changes in the associations with child mortality.[31] Most children remain residents of their migrating parent's origin household.[35] Further, the availability of government pensions allows children of migrants to stay with grandmothers who have resources to support them.

We showed that associations with kin and child mortality were moderated by parental status. We found a protective effect of related adults in one-parent households only in the period of widespread ART availability. The lack of a protective effect in the earlier HIV period likely reflects complex household dynamics given excess AIDS mortality in prime aged adults.[23 25] For instance, one-parent households during this time period were more likely to have experienced loss of a parent. Related adults may have also placed additional burden on the household in this configuration,[36] for instance if they were unwell themselves. With widespread ART availability, the likelihood of parental loss and unwell-related adults would both be reduced, which may reflect these adults being able to now provide caretaking and other roles that support vulnerable children.[16 37] Presence of related adults may also help by reducing resource strain when funds are directed to unwell household members and providing care for children when mothers become very ill.[2]

Our findings suggest heterogeneity in the configuration of households and their associations with child mortality between the two sites. Household diversity has changed over time, in part due to low marriage rates, high levels of labour migration and unemployment and high HIV prevalence.[38 39] Results from reviews on the role of kin on child mortality have shown between study heterogeneity in kin effects, due in part to different study designs and lack of consistent controls.[40 41] A strength of our study is the use of harmonised data and a unified statistical framework to examine differences in kin associations between the two sites. Our results provide evidence for the supportive role of kin, dependent on different ecological and epidemiological conditions that vary over time and between local contexts.

We acknowledge study limitations. First, our household variables only include matrilineal kin given the greater consistency over time of maternal identifiers. To create a harmonised data structure across the two sites, we also only identified household structure and composition based on memberships information. We therefore lack detailed information on the roles of matrilineal kin on child caretaking and nutrition, and how this may vary by parental status. Formative research is needed to understand the role of other household kin, particularly in one-parent households. Using household memberships also means that we cannot account for the role of kin who may provide child caretaking or support but are members of another household. Associations between household structure and composition may also be due to other unmeasured risk factors. We used multilevel modelling to account for shared mortality risk for children in the same household. Our results were also robust to controls for household SES and household head gender.

A key contribution of this study was to provide comparative, longitudinal evidence on associations between household structure and composition with child mortality from two rural South African populations both heavily burdened by HIV/AIDS. With the rollout of ART and rapid changes in SES,[23 25 29] these settings are continuing to undergo rapid demographic, epidemiological and social transitions. Further longitudinal research is needed to understand continued changes in living arrangements and the role of parents and kin in protecting the well-being of children.

**Author affiliations**
[1]School of Demography, The Australian National University, Canberra, Australian Capital Territory, Australia
[2]MRC/Wits Rural Public Health and Health Transitions Research Unit (Agincourt), School of Public Health, Faculty of Health Sciences, University of the Witwatersrand, Acornhoek, South Africa
[3]Africa Health Research Institute, Somkhele, South Africa
[4]DSI-MRC South African Population Research Infrastructure Network (SAPRIN), Durban, South Africa
[5]Department of Sociology, The Ohio State University, Columbus, Ohio, USA

**Contributors** BH and CK conceived the study. BH wrote the first draft and designed and completed the statistical analyses. CK prepared the data with the support of DG. KH and SJC provided overall guidance to the conduct of the study. CK, DG, KH and SJC revised the manuscript for important intellectual content and contributed to interpretation of the data. All authors read and approved the final manuscript. BH is the guarantor for the study.

**Funding** The authors received no specific funding for this study. The MRC/Wits Rural Public Health and Health Transitions Research Unit and Agincourt Health and Socio-Demographic Surveillance System, a node of the South African Population Research Infrastructure Network (SAPRIN), is supported by the Department of Science and Innovation, the University of the Witwatersrand and the Medical Research Council, South Africa, and previously the Wellcome Trust, UK (grants 058893/Z/99/A; 069683/Z/02/Z; 085477/Z/08/Z; 085477/B/08/Z). The Africa Health Research Institute is supported largely by Wellcome Trust (grant 201433/Z/16/Z) and Howard Hughes Medical Institute. The population surveillance, a node of the South African Population Research Infrastructure Network (SAPRIN), is funded by the Department of Science and Innovation and hosted by the South African Medical Research Council.

**Competing interests** None declared.

**Patient and public involvement** Patients and/or the public were not involved in the design, or conduct, or reporting, or dissemination plans of this research.

**Patient consent for publication** Not applicable.

**Ethics approval** Ethics approval for Agincourt Health and socio-Demographic Surveillance System was obtained from the Human Research Ethics Committee (Medical) of the University of the Witwatersrand, Johannesburg, South Africa (protocols M960720 and M110138). Ethics approval for AHRI was obtained from the Biomedical Research Ethics Committee (BREC) of the University of KwaZulu-Natal, Durban, South Africa (reference number BE169/15). Both sites obtain and document informed verbal consent at each census visit from the head of household or proxy adult respondent. This verbal consent process is standard across the INDEPTH Network of demographic surveillance sites (DSS) given the infeasibility of contacting every person in the DSS. The above ethics committees have continued to approve the verbal consenting process.

**Provenance and peer review** Not commissioned; externally peer reviewed.

**Data availability statement** Data are available upon reasonable reueqst. The data underlying this article will be shared on reasonable request to the corresponding author. The data underlying the results presented in the study are available from the Africa Health Research Institute (AHRI) Data Repository (https://data.ahri.org) for researchers who meet the criteria for access to confidential data and sign on the agreement according to the AHRI's policy for data sharing. Customised data extraction can be requested from AHRI data management service desk ( rdmservicedesk@ahri.org). Detailed documentation of the AHDSS data and an anonymised database containing data from 10% of the surveillance households are available for public access on the AHDSS website (http://www.agincourt.za). The AHDSS core demographic data are also routinely deposited for public access in the INDEPTH Network Data Repository (http://www.indepthishare.org/) and the South African Population Research Infrastructure Network (SAPRIN) Data Repository (http://saprindata.samrc.ac.za/index.php/catalog). Customised data extraction can be requested from Dr F Xavier Gómez-Olivé (F.Gomez-OliveCasas@wits.ac.za).

**ORCID iDs**
Brian Houle http://orcid.org/0000-0003-2157-3118
Dickman Gareta http://orcid.org/0000-0002-4004-5655

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
