## [Reviewer comments · BMJ Open]

ARTICLE DETAILS

TITLE (PROVISIONAL)	Household structure, composition, and child mortality in the unfolding antiretroviral therapy era in rural South Africa: Comparative evidence from population surveillance, 2000-2015
AUTHORS	Houle , Brian; Kabudula, Chodziwadziwa; Gareta, Dickman; Herbst, Kobus; Clark, Samuel J

VERSION 1 – REVIEW

REVIEWER	Li, Li-ming Peking University
REVIEW RETURNED	23-Dec-2022

GENERAL COMMENTS	Generally, this was a well-conducted study. I have 2 minor problems. 1. Please provide information on informed consent from participants or their parents. This was not described in the Ethics approval section. 2. In Figure 2, the authors used line chart, which might be inappropriate as the abscissa was different types of household structure.
---

REVIEWER	Wekesa, Paul Centre for Health Solutions-Kenya
REVIEW RETURNED	24-Dec-2022

GENERAL COMMENTS	I have enjoyed reading your interesting manuscript on household structure, composition, and child mortality in the ART era in rural South Africa. The manuscript is very clear, concise and presents sound evidence. A few comments to consider: 1. While protective factors are well presented and discussed, would the authors be interested in discussing factors that increase the odds of child mortality as well? 2. Do the authors have data on causes of child mortality in the two DSS sites for the period? Could this be described to provide an overview for the reader? 2. Did the authors have any clinical monitoring data for children on ART in the follow-up period?
---

VERSION 1 – AUTHOR RESPONSE

Reviewer: 1

Comments to the Author:

Generally, this was a well-conducted study. I have 2 minor problems.

1. Please provide information on informed consent from participants or their parents. This was not described in the Ethics approval section.

We thank the reviewer for this suggestion. We have added a paragraph on the informed consent process for each site in the Ethics approval section.

“Both sites obtain informed verbal consent at each census visit from the head of household or proxy adult respondent and this is documented in the data collection instruments. This verbal consent process is standard across the INDEPTH Network of DSSs given the infeasibility of contact every person in a DSS. The ethics committees have continued to approve the verbal consenting process.”

2. In Figure 2, the authors used line chart, which might be inappropriate as the abscissa was different types of household structure.

We have updated Figure 2 to omit the lines so that only the estimates and 95% confidence intervals are shown for the different types of household structure.

Reviewer: 2

Comments to the Author:

I have enjoyed reading your interesting manuscript on household structure, composition, and child mortality in the ART era in rural South Africa. The manuscript is very clear, concise and presents sound evidence. A few comments to consider:

1. While protective factors are well presented and discussed, would the authors be interested in discussing factors that increase the odds of child mortality as well?

We thank the reviewer for this suggestion. We have updated the presentation in the results to also discuss which factors increase the odds of child mortality. We do this when introducing the first model to highlight what other factors put children at risk.

“The results from the full household structure model are presented in online supplemental table 1. An interaction between DSS and time period ($p < 0.001$), DSS and household structure ($p = 0.001$), and time and household structure ($p = 0.004$) significantly improved model fit. A multi-level model including a mother random intercept improved model fit according to the Bayesian Information Criterion ($\Delta BIC = 156$) and resulted in the final model. Twins, boys, younger children, and children in households with two or more other children have higher odds of mortality.”

2. Do the authors have data on causes of child mortality in the two DSS sites for the period? Could this be described to provide an overview for the reader?

We agree with the reviewer that information on causes of child mortality would be useful to provide an overview for the reader. Each site conducts a verbal autopsy for deaths that are identified during a census follow-up. We have used that information to provide a comparison over time for each site on causes of death (new online supplemental figure 2). We added a brief discussion of this to the methods and results:

Methods updates

“Each site conducts a verbal autopsy (VA) for individuals who died during surveillance rounds. A trained fieldworker or nurse uses a standardized VA instrument to interview the closest living relative of the decedent to record signs and symptoms experienced before their death.”

“We used InterVA-5 to assign causes of death.²⁸ InterVA is a model that assigns up to three causes of death based on the VA interview data – we used the cause of death with the largest likelihood for each complete VA interview. This allowed us to compare how the distribution of causes of death changed over time within each DSS.”

Results updates

“We first describe under-five mortality patterns by time and DSS. Out of a total of 101,105 children, 3,603 died between 2000-2015. Online supplemental figure 1 shows the mortality rate per 1,000 for children under 5 by year and DSS. Mortality began decreasing at AHRI in 2004 and AHDSS in 2009. Online supplemental figure 2 shows the distribution of child cause of death over the two time periods by DSS. For both AHRI and AHDSS, the share of deaths due to HIV/AIDS and TB declined in 2008-2015 compared to 2000-2007, while the share of deaths due to respiratory infections has increased over time.”

Online supplemental figure 2

Supplemental Figure 2. Distribution of child causes of death over time, Agincourt Health and Demographic Surveillance System (AHDSS) and Africa Health Research Institute (AHRI), South Africa 2000-2007 and 2008-2015. Child causes of death classified by InterVA-5 based on VA.

3. Did the authors have any clinical monitoring data for children on ART in the follow-up period?

The reviewer raises an excellent point. Clinical data are not routinely collected by both sites during the census updates to develop a representative set of data. While there are efforts at different stages at both sites to link public clinic data to the census, linking with children is more difficult and we lack the adequate human subjects clearance to do this with our data set.

VERSION 2 – REVIEW

REVIEWER	Wekesa, Paul Centre for Health Solutions-Kenya
REVIEW RETURNED	14-Feb-2023
GENERAL COMMENTS	Thank you for working on the comments suggested in the previous review.